# Tertiary Structures of Haseki Tick Virus Nonstructural Proteins Are Similar to Those of *Orthoflaviviruses*

**DOI:** 10.3390/ijms252413654

**Published:** 2024-12-20

**Authors:** Anastasia Gladysheva, Irina Osinkina, Nikita Radchenko, Daria Alkhireenko, Alexander Agafonov

**Affiliations:** 1State Research Center of Virology and Biotechnology “Vector”, 630559 Kol’tsovo, Russia; osinkina_ia@vector.nsc.ru (I.O.); radchenko_ns@vector.nsc.ru (N.R.); alhireenko_da@vector.nsc.ru (D.A.); agafonov@vector.nsc.ru (A.A.); 2Natural Sciences Department, Novosibirsk State University, 630090 Novosibirsk, Russia

**Keywords:** ixodid ticks, tick-borne infection, RNA viruses, *Flaviviridae*, orthoflavi-like viruses, viral proteins, protein structure, AlphaFold 3

## Abstract

Currently, a large number of novel tick-borne viruses potentially pathogenic to humans are discovered. Studying many of them by classical methods of virology is difficult due to the absence of live viral particles or a sufficient amount of their genetic material. In this case, the use of modern methods of bioinformatics and synthetic and structural biology can help. Haseki tick virus (HSTV) is a recently discovered tick-borne unclassified ssRNA(+) virus. HSTV-positive patients experienced fever and an elevated temperature. However, at the moment, there is no information on the tertiary structure and functions of its proteins. In this work, we used AlphaFold 3 and other bioinformatic tools for the annotation of HSTV nonstructural proteins, based on the principle that the tertiary structure of a protein is inextricably linked with its molecular function. We were the first to obtain models of tertiary structures and describe the putative functions of HSTV nonstructural proteins (NS3 helicase, NS3 protease, NS5 RNA-dependent RNA-polymerase, and NS5 methyltransferase), which play a key role in viral genome replication. Our results may help in further taxonomic identification of HSTV and the development of direct-acting antiviral drugs, POC tests, and vaccines.

## 1. Introduction

Preventive measures against viral threats represent one of the most important issues of biological security. The high virulence and variability of many dangerous human viruses as well as their ubiquitous distribution make investigation of novel viruses an important stage in protection from infectious threats.

According to the International Committee on Taxonomy of Viruses (ICTV) report, 11,273 viral species, of which more than 3000 are pathogenic to humans, have been identified by 2023 [1]. However, according to virologists’ suggestions, the number of virus species not studied by humans may exceed several millions, many of which may be pathogenic to humans. There are entire eras in human history when viral pandemics and epidemics led to mass infections and many deaths, as was the case with the COVID-19 pandemic [2]. According to annual WHO reports, viral infections account currently for 60 to 75% of infectious diseases. Furthermore, viruses exhibit sufficiently high genetic variability to overcome interspecies barriers or accumulate mutations that significantly increase their virulence. Therefore, investigation of novel, previously unidentified viruses is a key task of preventive measures against viral threats.

Ticks are the vectors of over 90% of vector-borne diseases and represent a growing public health problem. As global climate conditions change and humans encroach on previously untouched natural areas, tick distribution is expected to expand. This may result in longer periods of tick activity, increasing the likelihood of transmission of tick-borne viral infections, such as *Flaviviridae* infections, to humans as people and animals are exposed to potential tick bites for longer periods. In recent years, many *Flaviviridae* like tick-borne viruses potentially pathogenic to humans have been identified using NGS: Jingmen tick virus, Alongshan virus, Haseki tick virus, etc. Many of them have not been annotated beyond the polyprotein level.

Haseki tick virus (HSTV) was first discovered in a retrospective study of blood sera from tick-bitten patients and *Ixodes persulcatus* ticks from various regions of the Russian Federation in 2019. There are reports of identification of viruses closely related to HSTV in Georgia and Poland [3]. The HSTV genome is represented by a single-stranded positive-sense RNA of approximately 16,000 nucleotides in size that encodes an approximately 5100 amino acid polyprotein flanked by 5′- and 3′-untranslated regions. On the basis of the genome structure, this virus was initially classified as an unclassified orthoflavi-like virus of the *Flaviviridae* family. Many members of this family cause serious diseases in humans. HSTV-infected patients experienced fever and elevated temperature [4]. However, the ability of HSTV to cause symptomatic infection in humans requires further investigation. HSTV is also closely related to other recently identified unclassified pestiviruses: Bole tick virus 4 (China, Thailand, Slovakia, Romania, and Kenya), Trinbago virus (China and Trinidad and Tobago), and *Dermacentor reticulatus* pestivirus-like virus 1 (Croatia, Georgia, and Poland), which have been detected in *Dermacentor reticulatus*, *Rhipicephalus sanguineus*, *Rhipicephalus turanicus*, *Hyalomma punctata*, *Hyalomma truncatum*, *Hyalomma rufipes,* and *Hyalomma dromedarii* ticks [3,5,6,7,8,9]. In addition, Bole tick virus 4 was detected in skin and serum samples from tick-bitten patients presenting with both mild and severe symptoms [10]. Despite the ubiquitous distribution of HSTV-related viruses, information on the structure and function of their viral proteins is currently lacking. Furthermore, it remains unclear whether HSTV is an *Orthoflavivirus*, as it was originally identified, or a *Pestivirus* such as HSTV-related viruses. This poses a serious challenge to understanding the mechanisms and evolution of HSTV viral infection, precludes taxonomic identification of HSTV, and establishment of continuous monitoring. Given the widespread prevalence of HSTV-related viruses, including in patients after tick bites, more attention should be paid to targeted studies of HSTV proteins’ structure for the development of treatment in the future.

We hypothesize that the tertiary structure of HSTV nonstructural proteins is similar to those of *Orthoflaviviruses*, which have medical and economic significance. This may indicate similar properties of these viruses. But many newly discovered tick-borne viruses cannot be studied using classical virology techniques due to the laborious culture of novel viruses or the lack of a sufficient amount of their genetic material. In this case, modern synthetic and structural biology techniques enable the artificial production of recombinant proteins identical to natural viral ones and the investigation of their structure and properties. However, X-ray structural analysis cannot be often used for proteins of novel viruses due to difficulties in producing protein crystals of sufficient size and requires additional approaches to phase restoration. A promising technique to study the structure and functions of novel unique viral proteins is a recently announced AlphaFold 3 neural network [11].

In this study, we used the AlphaFold 3 neural network and other bioinformatics methods to identify key putative HSTV nonstructural proteins in the polyprotein, which lacked annotated sequence homologs.

## 2. Results

### 2.1. Determination of Localization of Putative Haseki Tick Virus Nonstructural Proteins

Nonstructural (NS) proteins of *Flaviviridae* family members interact with several host factors to form a membrane-bound replication complex where viral RNA is synthesized. Some NS proteins are transmembrane and anchored to the membrane of the endoplasmic reticulum. Other NS proteins are localized to the membrane via membrane-associated regions or viral protein cofactors [12]. Because protein structure is commonly regarded to be conserved and to dictate function, we used a structural approach to search for NS proteins in the C-terminal region of the HSTV polyprotein (1272–5104 a.a.), comparing NS protein structures of *Orthoflavivirus* genus (NS2A, NS2B, NS3, NS4A, NS4B, and NS5), *Hepacivirus* genus (NS2, NS3, NS4A, NS4B, NS5A, and NS5B), and *Pestivirus* genus (NS2, NS3, NS4A, NS4B, NS5A, and NS5B) viruses available in PDB. However, the possible structural divergence of the HSTV proteins with proteins of known *Flaviviridae* members may indicate differences in the evolution rate of these viruses and their ability to adapt to new natural and climatic conditions. There are examples when minor structural divergence of viral proteins of the *Orthoflavivirus* genus led to the impossibility of developing vaccines and antiviral drugs, while yellow fever, Japanese encephalitis, and tick-borne encephalitis vaccines exist.

#### 2.1.1. Search for Putative HSTV Nonstructural Transmembrane Proteins

In the nonstructural region of the HSTV polyprotein, two transmembrane proteins (NSTR1, NSTR2) were found at positions 1272–1601 and 2833–2855, which may be associated with NS2A/NS2B and NS4A/NS4B proteins of *Flaviviridae* family members (Figure 1 and Appendix A). HSTV NSTR1 consists of four transmembrane domains of 21 a.a. (NSTR1-D1), 17 a.a. (NSTR1-D2), 24 a.a. (NSTR1-D3), and 24 a.a. (NSTR1-D4) in length and one 127 a.a. cytoplasmic domain (~14 kDa). Between NSTR1-D2 and NSTR1-D3, there is a 24 a.a. membrane-localized domain. In *Flaviviridae* family members, the NS2A protein is reported to consist of five transmembrane domains and be about 220 a.a. in length, in contrast to the four transmembrane domains in the HSTV polyprotein. The putative HSTV NSTR1 protein is preceded by a 17 a.a. membrane-bound region, the role of which we did not investigate. A complex between NSTR1-D4 and the NSTR1 cytoplasmic domain may be a prototype of the NS2B protein that is a cofactor of the NS3 serine protease of *Orthoflavivirus* genus members. HSTV NSTR2 consists of four transmembrane domains of 23 a.a. (NSTR2-D1), 26 a.a. (NSTR2-D2), 18 a.a. (NSTR2-D3), and 20 a.a. (NSTR2-D4) in length, which are located one after another. In *Flaviviridae* family members, three transmembrane domains in the NS4A protein and four transmembrane domains in the NS4B protein have been reported, which is inconsistent with the data on four transmembrane domains in the HSTV sequence. We were unable to predict the tertiary structures of HSTV NSTR1 and HSTV NSTR2 due to the lack of structural data on NS2A and NS4A/NS4B proteins of *Flaviviridae* family members in PDB. In addition, at the HSTV polyprotein C-terminus (4910–5104 a.a.), there are four 20 a.a., 23 a.a., 34 a.a., and 20 a.a. transmembrane domains, whose function is unknown (Figure 1 and Appendix A).

#### 2.1.2. Search for Putative HSTV Nonstructural Cytoplasmic Proteins

Putative NS3 protein

All four *Flaviviridae* genera (*Orthoflavivirus*, *Hepacivirus*, *Pegivirus,* and *Pestivirus*) have the ~70 kDa NS3 protein in their genome structure. NS3 consists of two domains: the N-terminal domain of serine protease, which is a key protease in post-translational cleavage of the viral polyprotein, and the C-terminal domain of helicase, which plays an important role in the unwinding of viral RNA duplexes during replication [13].

Putative NS3 was found in the HSTV polyprotein at positions 1598–2291 (Figure 1). The size of HSTV NS3 was 694 a.a. (~77 kDa). Putative HSTV NS3 consists of HSTV serine NS3 protease (NS3pro) of 204 a.a. in length (~22 kDa), located in the polyprotein at positions 1598–1801, and HSTV NS3 helicase (NS3hel) of 490 a.a. in length (~55 kDa), located in the polyprotein at positions 1802–2291. The sizes of HSTV NS3pro and HSTV NS3hel are consistent with those of similar viral proteins of *Flaviviridae* family members. The HSTV NS3pro amino acid sequence is most similar to that of Dengue virus 2 NS3pro (UniProtKB accession number: Q91H74) and Zika virus NS3pro (UniProtKB accession number: H8XX12), but their amino acid sequence identity is 23% and 21%, respectively. The identity of the HSTV NS3pro amino acid sequence with that of other *Flaviviridae* family members is not more than 13% for *Hepacivirus* and 8% for *Pestivirus*. The HSTV NS3hel amino acid sequence is most similar to that of classical swine fever virus NS3hel (UniProtKB accession number: Q5U8X5), but their amino acid sequence identity is 28%. Homology between HSTV NS3hel and those from *Orthoflavivirus* (UniProtKB accession number: P14336) and *Hepacivirus* (UniProtKB accession number: Q9WMX2) is not more than 21%.

Putative Methyltransferase and RNA-dependent RNA polymerase

*Orthoflavivirus* genus members possess a ~103 kDa NS5 protein consisting of two functional domains with methyltransferase (NS5MTase) and RNA-dependent RNA polymerase (NS5RdRp) activities, whereas *Pegivirus*, *Pestivirus,* and *Hepacivirus* lack NS5MTase, but have the phosphoprotein NS5A and NS5B that functions as RdRp [14]. NS5MTase is responsible for the formation of the RNA cap attached to the 5′-end of the viral RNA. NS5 MTase transfers a methyl group from an S-adenosyl-L-methionine donor to N-7-guanine and ribose 2′-OH of the first RNA nucleotide to form an S-adenosyl-L-homocysteine by-product. NS5A is a multifunctional phosphoprotein consisting of three domains (DI, DII, and DIII). The DI and DII domains are involved in genome replication, whereas DIII plays a role in virus assembly [15]. NS5RdRp synthesizes viral RNA [12].

In the HSTV polyprotein, putative HSTV NS5MTase was found at positions 3445–3676 a.a., and putative HSTV NS5RdRp was found at positions 3999–4709 a.a. (Figure 1). The NS5A phosphoprotein was not detected in HSTV. The size of HSTV NS5MTase and HSTV NS5RdRp was 231 a.a. (~26 kDa) and 711 a.a. (~82 kDa), respectively. Between HSTV NS5MTase and HSTV NS5RdRp, there is a 323 a.a. (~36 kDa) protein domain with unknown properties (NS5-X). This protein domain remains folded only when complexed with HSTV NS5MTase and HSTV NS5RdRp (Figure 1). The detected putative HSTV NS5MTase has no amino acid sequence homologs. The level of identity of HSTV NS5MTase with NS5MTase of *Orthoflavivirus* members is ~5%. The HSTV NS5RdRp amino acid sequence is most similar to that of the bovine viral diarrhea virus (UniProtKB accession number: Q96662) and classical swine fever virus (UniProtKB accession number: Q5U8X5), but their amino acid sequence identity is only 23% and 22%, respectively. Homology between HSTV NS5RdRp and that of *Orthoflavivirus* (UniProtKB accession number: P14336) and *Hepacivirus* (UniProtKB accession number: Q9WMX2) members is not more than 17%.

To search for potential cleavage sites of the HSTV polyprotein, the amino acid sequence of the N-terminus of HSTV NS3 was aligned with the N-termini of NS3: CSFV (UniProtKB accession number: P21530), ZIKV (UniProtKB accession number: Q32ZE1), YFV (UniProtKB accession number: P03314), TBEV (UniProtKB accession number: Q01299), HCV (UniProtKB accession number: P26664); the C-terminus of HSTV NS3 was aligned with the C-termini of NS3: CSFV (UniProtKB accession number: P21530), ZIKV (UniProtKB accession number: Q32ZE1), YFV (UniProtKB accession number: P03314), TBEV (UniProt: Q01299), HCV (UniProt: P26664); the N-terminus of HSTV NS5Mtase was aligned with the N-termini of NS5: ZIKV (UniProtKB accession number: Q32ZE1), YFV (UniProtKB accession number: P03314), TBEV (UniProtKB accession number: Q01299); the N-terminus of HSTV NS5RdRp was aligned with the N-termini of NS5b (RdRp): CSFV (UniProtKB accession number: P21530), HCV (UniProtKB accession number: P26664); the C-terminus of HSTV NS3 and the N-terminus of HSTV NSTR2 (putative NS4A analog) was aligned with the C-termini of NS3 and the N-termini of NS4A: CSFV (UniProtKB accession number: P21530), ZIKV (UniProtKB accession number: Q32ZE1), YFV (UniProtKB accession number: P03314), TBEV (UniProtKB accession number: Q01299), HCV (UniProtKB accession number: P26664); the C-terminus of HSTV NSTR2 (putative NS4B analogue) and the N-terminus of HSTV NS5Mtase were aligned with the C-termini of NS4B and the N-termini of NS5: ZIKV (UniProtKB accession number: Q32ZE1), YFV (UniProtKB accession number: P03314), TBEV (UniProtKB accession number: Q01299). We were unable to identify potential cleavage sites of the HSTV polyprotein based on sequence alignment (Appendix A). This may be due to the different specificities of the HSTV proteases and those of the *Flaviviridae* family. However, based on literature data, several potential HSTV proteolysis sites were identified (Appendix A).

### 2.2. Tertiary Structure Models of Putative Haseki Tick Virus Nonstructural Proteins

#### 2.2.1. Putative HSTV NS3 Protein

We generated a spatial structure model of putative HSTV NS3 with pLDDT = 73.59 (Appendix A). Pairwise alignment of the full-length HSTV NS3 with the NS3 crystal structures of *Flaviviridae* family members revealed a low level of structural similarity (Figure 2). The highest TM score = 0.62 was found for Dengue virus 4 NS3 (PDB ID: 2VBC) (Appendix A). HSTV NS3pro and HSTV NS3hel are interconnected by a 13 a.a. flexible linker (Appendix A). This leads to instability of the spatial arrangement of HSTV NS3pro relative to HSTV NS3hel and, thus, to a low TM score. Therefore, we further analyzed the spatial structures of HSTV NS3pro and HSTV NS3hel separately.

Putative HSTV NS3 protease

We could generate an HSTV NS3pro model with a high model confidence level (pLDDT = 70.40) (Appendix A). These data indicate that the HSTV NS3pro model may be used for functional analysis. The structure of NS3 protease was found to display the highest level of topological similarity to that of analogous proteins of *Orthoflavivirus* members, despite the fact that their amino acid sequence identity was less than 30% (Table 1 and Appendix A). The TM score of HSTV NS3pro with DENV NS3pro and ZIKV NS3pro ranged from 0.70 to 0.79 and from 0.69 to 0.75, respectively, whereas the TM score of HSTV NS3pro with HCV NS3pro varied from 0.60 to 0.69 (Appendix A).

The tertiary structure of HSTV NS3pro consists of two spatial domains (D1 and D2) composed of β-barrels. HSTV NS3pro D1 consists of seven antiparallel β-sheets (β1, β2, β3, β4, β5, β5, and β7), and HSTV NS3pro D2 consists of eight antiparallel β-sheets (β1′, β2′, β3′, β4′, β5′, β6′, β7′, and β8′). A small 5 a.a. α-helix is present between β3 and β4 in HSTV NS3pro D1, which is completely consistent with the tertiary structures of NS3pro of *Flaviviridae* family members from PDB. HSTV NS3pro D1 and D2 are connected by a 12 a.a. flexible linker (Appendix A). There are additional 9 and 13 a.a. insertions in HSTV NS3pro at amino acid positions 37–46 and 110–123, respectively. These insertions increase the sizes of β4, β5, β10, and β11 sheets in HSTV NS3pro and partially form unstructured free loops.

We identified the active site of HSTV NS3pro (Figure 3a) similar to that of NS3pro in *Orthoflavivirus* genus members. The NS3pro active site in *Orthoflavivirus* genus members is formed by a catalytic His51–Asp75–Ser135(Gly133) triad, where these amino acids act as a nucleophile, base, and acid, respectively, upon peptide bond cleavage. Gly133 is not directly involved in the catalytic site, but, together with Ser135, forms a structural fold that acts as an anion pocket during nucleophilic attack. Topologically, the HSTV NS3pro active site completely coincides with that of NS3pro in *Orthoflavivirus* genus members and is formed by a catalytic His55–Asp88–Ser163(Gly161) triad. Differences in the coordinates are caused by additional amino acid insertions in the HSTV NS3pro structure. The HSTV NS3pro active site is negatively charged and forms a charged pocket surrounded by hydrophobic regions (Figure 3b,c). In *Orthoflaviviruses*, NS3pro functions in a complex with NS2B that acts as a protease cofactor. In the complex, NS2B usually stabilizes the NS3pro active site and acts as an electron donor in peptide bond cleavage [16]. Simulation of the tertiary structure of a complex between HSTV NS3pro and the HSTV NSTR1 cytoplasmic domain revealed similarity to the topology of an NS2B–NS3pro complex in *Orthoflavivirus* genus members (Figure 3d). Upon simulation of the complex, the HSTV NS3pro cytoplasmic domain folds to a structure composed of three β-sheets interacting with the HSTV NS3pro surface (Appendix A).

Putative HSTV NS3 helicase

A tertiary structure model of HSTV NS3hel with pLDDT = 75.80 was generated using AlphaFold 3 (Appendix A). These data indicate that the HSTV NS3hel model may be used for functional analysis. HSTV NS3hel was found to have the highest topological similarity to *Hepacivirus hominis* NS3hel (TM score: 0.66), despite low amino acid sequence identity (less than 30%). The TM score of HSTV NS3hel with Dengue virus 4 NS3hel and Zika virus NS3hel ranged from 0.60 to 0.63 and from 0.61 to 0.63, respectively, whereas the TM score of HSTV NS3hel with tick-borne encephalitis virus NS3hel and classical swine fever virus NS3hel was 0.62 and 0.63, respectively (Table 1 and Appendix A, Appendix A).

The HSTV NS3hel structure has the shape of a flattened pyramid that can be divided into three domains: N-terminal domains 1 and 2 (D1, residues 1802–1962; D2, residues 1963–2131) and C-terminal domain 3 (D3, residues 2132–2291) (Appendix A and Figure 4). HSTV NS3hel D1 and D2 are tandem, conserved RecA-like domains with an α/β fold. HSTV NS3hel D1 consists of six parallel β-sheets (β1, β2, β3, β5, β6, and β7) and one antiparallel β-sheet (β4), which are sandwiched by seven α-helices (α1–α7). Whereas HSTV NS3hel D2 contains six parallel β-sheets (β1′, β4′, β5′, β6′, β7′, and β9′) and two antiparallel β-sheets (β2′ and β3′) sandwiched by four α-helices (α1′–α4′) (Appendix A). In addition, a β-hairpin consisting of a pair of antiparallel β-sheets (β8A’–β8B’) protrudes from the HSTV NS3hel D2 domain and interacts with the HSTV NS3hel D3 domain. In particular, the β-hairpin is packed opposite a hydrophobic region formed by α1′′, α2′′, and the N-terminus of α5′′ of the HSTV NS3hel D3 domain (Figure 5a). The structure of the HSTV NS3hel D3 domain is very limitedly similar to that of NS3hel D3 in *Flaviviridae* family members, which is consistent with the high variability of D3 in viral helicases (Table 1 and Appendix A). At polyprotein positions 1869–1887, HSTV NS3hel has an 18 a.a. insertion that forms a β3-sheet.

All motifs (I or Walker A, Ia, II or Walker B, III, IV, IVa, V, VI) characteristic of superfamily 2 helicases were found in HSTV NS3hel (Figure 4 and Appendix A) [13]. These motifs are located in a cleft between HSTV NS3hel D1 and D2 domains. The Walker A motif of HSTV NS3hel consists of a residue triad Gly1827, Lys1828, and Ser2079 and, together with the Walker B motif of HSTV NS3hel, consisting of residues Asp1930, Glu1931, and His1933, is responsible for binding to nucleotide triphosphatase (NTPase) and coordination of Mg^2+^ [17]. Motif III of HSTV NS3hel is composed of residues Thr1958 and Thr1960 and, together with Gln2108 of HSTV NS3hel motif VI, forms an exit channel for inorganic phosphate produced in hydrolysis [18]. Of critical importance for the NTPase activity is motif VI which is formed in HSTV NS3hel by residues Arg2109, Arg2110, Arg2112, and Arg2115 [13].

In the model of the HSTV NS3hel structure, a positively charged tunnel is clearly identified along the boundary of HSTV NS3hel D3, which directly interacts with HSTV NS3hel D1 and D2 (Figure 5b). The tunnel is lined with positively charged amino acids and is wide enough to accommodate a single-stranded nucleic acid passing through HSTV NS3hel D2 to D1. The positively charged residues, most of which belong to HSTV NS3hel D1 and D2, presumably stabilize the nucleic acid sugar–phosphate backbone [13]. To assess the interaction of HSTV NS3hel with RNA and ATP, we built a model of an HSTV NS3hel–ATP–50-nucleotide random RNA complex in AlphaFold 3 (Figure 5c). The model shows that the RNA is indeed coordinated in the positively charged tunnel along the HSTV NS3hel D3 boundary.

#### 2.2.2. Putative HSTV NS5 RNA-Dependent RNA Polymerase

A tertiary structure model of HSTV NS5RdRp had a high pLDDT = 87.41, despite the lack of amino acid sequence homologs (Appendix A). The highest TM score = 0.72 was found for HSTV NS5RdRp with Dengue virus 2 NS5RdRp (PDB ID: 7XD8) and Zika virus NS5RdRp (PDB ID: 5U0C). The structural similarity of the HSTV NS5RdRp to the NS5RdRp of tick-borne encephalitis virus was 0.69. The structural similarity of HSTV NS5RdRp to NS5RdRp of *Orthoflavivirus* genus members and NS5RdRp of *Pestivirus* genus members was 0.66 to 0.72 and 0.66, respectively, whereas the TM score for HSTV NS5RdRp with NS5RdRp of *Hepacivirus hominis* species was 0.63 (Table 2 and Appendix A, Appendix A).

The spatial structure model of HSTV NS5RdRp adopts a right-hand shape with palm, finger, and thumb domains surrounding the active site, which is similar to that in all viral RdRps (Appendix A). The HSTV NS5RdRp finger domain consists of two regions: The first region is located in helices α1–8 and sheets β1–5 (residues 3999–4214), and the second region is located in helices α11–12 and sheets β7–8 (residues 4273–4316). The HSTV NS5RdRp palm domain is spread throughout HSTV NS5RdRp. The palm domain forms the RdRp core and is composed of helices α9–10 (residues 4215–4272), α15–16 (residues 4317–4453), and α13–14 as well as beta sheets β9–15 (Appendix A). The HSTV NS5RdRp thumb domain is located at the C-terminal side of RdRp, consists of helices α15–27 (residues 4454–4709), and, together with the HSTV NS5RdRp palm domain, forms a dsRNA interaction channel (Appendix A).

In the HSTV NS5RdRp palm domain, we identified all the motifs characteristic of viral RdRps, which were located in positions of the spatial structure similar to those in other *Flaviviridae* family members (Appendix A). Given their spatial and amino acid similarity, we suggest that the identified HSTV NS5RdRp motifs perform functions similar to those in other *Flaviviridae* RdRps. HSTV NS5RdRp motifs D, E, and G are highly variable, but their spatial arrangement around the polymerase active site remains conserved. HSTV NS5RdRp motifs A–E are located within the most conserved palm domain, and HSTV NS5RdRp motifs F and G are located in the finger domain (Figure 6). HSTV NS5RdRp motifs A and C contain conserved aspartic acid residues (Asp4260, Asp4265, Asp4363, and Asp4364) that play a key role in the catalytic center activity, coordinating metal ions [19]. The HSTV NS5RdRp motif B includes conserved Ser4320 and Gly4321. Ser4320 is specific for RdRps and forms hydrogen bonds with the 2′-hydroxyl group of ribose and Asp4265 of the HSTV NS5RdRp motif A. The HSTV NS5RdRp motif D starts after α14, mainly consists of unstructured loops, and forms an antiparallel β-structure with the HSTV NS5RdRp motif A. The HSTV NS5RdRp motif E interacts with the HSTV NS5RdRp motif C, and they both stabilize the de novo synthesized RNA product [20]. In the *Flaviviridae* family, the HSTV NS5RdRp motif E consists of a β-hairpin (β14–β15) located between the HSTV NS5RdRp palm and thumb domains [21,22]. However, the HSTV NS5RdRp motif E contains a 20 a.a. insertion that forms a free loop. In addition, HSTV NS5RdRp exhibits two additional α-helices (α21 and α22) extending from a putative priming loop that is important for the enzymatic activity of RdRp because flavivirus HSTV NS5RdRps belong to primer-independent polymerases [23].

#### 2.2.3. Putative HSTV NS5 Methyltransferase

We modeled the structure of an HSTV polyprotein portion between putative HSTV NSTR2 and HSTV NS5RdRp (3445–4000 a.a.) to search for either the NS5A protein, which is present in *Pestivirus*, *Hepacivirus,* and *Pegivirus* members, or the NS5 methyltransferase domain, which is present in *Orthoflavivirus*. The generated model had a low pLDDT = 36.76 and a TM score of <0.30 (respective to known *Flaviviridae* NS5A structures) (Appendix A). The protein model contained a highly structured and highly reliable N-terminal domain with a pLDDT = 71.10 and an extremely low pLDDT = 27.33 for an unstructured C-terminal domain. The HSTV N-terminal domain (Appendix A) had the highest structural similarity (TM score = 0.77) to methyltransferase from *Pyrococcus horikoshii* (PDB ID: 1WY7). There was also similarity to methyltransferases of *Orthoflavivirus* members for which the TM score ranged from 0.60 to 0.49. These data indicate that the modeled protein is HSTV methyltransferase (HSTV NS5MTase), despite the unusually distant location from NS5RdRp. Interestingly, the HSTV NS5MTase structure is highly identical to that in a recently discovered Alongshan virus (PDB ID: 8GY4) (Table 2 and Appendix A, Figure 7a,b and Appendix A).

The tertiary structure of HSTV NS5MTase folds into a classical α/β/α structure, where the central β-layer is surrounded by α-helices. The central β-layer consists of seven β-sheets (β1–β7) and contacts eight α-helices (α1–α8). The secondary structure of HSTV NS5MTase is as follows: α1–α2–α3–β1–α4–β2–α5–β3–α6–β4–α7–β5–α–β6–β7 (Figure 7c,d and Appendix A). Structural simulation of a complex of HSTV NS5MTase, HSTV NS5RdRp, and a protein between them (HSTV NS5-X) yields structured HSTV NS5-X (Figure 8a and Appendix A). The tertiary structure of HSTV NS5-X exhibits eight α-helices and eleven β-sheets. Electrostatic potential analysis of the HSTV NS5MTase–NS5-X–NS5RdRp complex reveals that HSTV NS5-X interacts with HSTV NS5MTase to form a positively charged pocket (Figure 8b). Probably, HSTV NS5-X promotes stabilization of the de novo synthesized HSTV RNA and plays an auxiliary role in HSTV RNA capping.

## 3. Discussion

Hidden Markov model profiling is a common procedure for functional annotation of novel viral genes in metagenomes [24]. However, this procedure is not applicable to genomes at less than 30% amino acid identity compared with that of annotated genomes. Thus, numerous protein sequences remain functionally unannotated and unclassified. The introduction of AlphaFold 2 in 2020 and then AlphaFold 3 in May 2024 marked the birth of a new era of protein folds and virome functions due to the application of deep learning and artificial intelligence algorithms in viral protein structure prediction [11]. Both multilayer neural networks use the primary and tertiary structures of proteins as input to model unknown structures. However, AlphaFold 3, unlike AlphaFold 2, is able to predict the 3D structure of biomolecular complexes of proteins, nucleic acids, and their ligands based solely on their linear sequences. This is a significant step toward understanding biomolecular interactions. In 2024, Hassabis and Jumper were awarded the Nobel Prize for developing artificial intelligence models to solve a problem in structural biology (https://www.nature.com/collections/edjcfdihdi) (accessed on 9 October 2024).

In this study, we used AlphaFold 3, based on the principle that the tertiary structure of a protein is inextricably linked to its molecular function [25,26]. Using AlphaFold 3, we were able for the first time to annotate and model tertiary structures and characterize putative functions of NS proteins: NS3 helicase, NS3 protease, NS5 methyltransferase, and NS5 RNA-dependent RNA polymerase of the recently discovered Haseki tick virus. In *Flaviviridae* members, the processes involved in virion morphogenesis are not fully understood, but interactions between structural and nonstructural proteins are known to be of critical importance [27]. NS3 and NS5 proteins play a key role in viral genome replication and may be targets for the development of direct-acting antiviral drugs [28,29]. In addition, we determined the localization of two HSTV membrane proteins (NSTR1 and NSTR2) that are putatively associated with NS2A/NS2B and NS4A/NS4B proteins that are viral cofactors and play an important role in viral RNA accumulation.

The primary structures of HSTV NS proteins lack homologs, so their identification in the HSTV polyprotein was difficult using homolog analysis methods. Despite a limited number of viral protein structures in the PDB and AlphaFold databases, the AlphaFold 3 neural network coped with the task of predicting HSTV NS proteins. The amino acid sequence sizes and molecular weights of the identified HSTV NS proteins are consistent with those of analogous proteins in *Flaviviridae* family viruses, whereas the HSTV genome size is approximately 1.5-fold larger. All the generated tertiary structures of HSTV proteins have a TM score of >0.5 compared with the tertiary structures of *Flaviviridae* viruses. This means that the protein structures have an approximately similar fold, and functional annotation of the proteins based on their structure is reasonable [30]. The tertiary structures of HSTV NS3hel, HSTV NS3pro, and HSTV NS5RdRp are most similar to those of Dengue viruses’ proteins. Every year, millions of people are infected with Dengue viruses through the bites of infected female *Aedes* mosquitoes. Licensed vaccines against dengue fever exist, but their limited availability in various countries makes it impossible to effectively protect people traveling to endemic countries [31]. The structural similarity of HSTV NS and Dengue viruses NS proteins may indicate the taxonomic unity of HSTV with *Orthoflavivirus* genus viruses. Whereas other unannotated viruses closely related to HSTV, such as Bole Tick Virus 4, Trinbago virus, and *Dermacentor reticulatus* pestivirus-like virus 1, were previously phylogenetically clustered as pesti-like viruses [3,5,6,7,8,9].

NS3 and NS5 of *Flaviviridae* members form a multi-enzyme protein complex that is primarily involved in the synthesis of positive- and negative-sense viral RNA and its capping [32].

NS3 consists of NS3 helicase and NS3 protease [13]. In pestiviruses and hepaciviruses, the cleavage between NS2 and NS3 is catalyzed by NS2, and among NS3, NS4A, NS4B, NS5A, and NS5B is catalyzed by NS3 protease. Whereas in *Orthoflaviviruses*, NS3 protease acts together with NS2B. NS2B of *Orthoflavivirus* genus members is a small protein consisting of two domains. The N-terminal domain of NS2B is transmembrane and is involved in the stabilization of the tertiary structure of NS3 protease. The soluble domain of NS2B is a cofactor of NS3 protease and forms the NS2B–NS3pro complex [33]. HSTV NS3 contains HSTV NS3pro and HSTV NS3hel domains, typical of *Flaviviridae*, connected by a flexible linker. HSTV NS3pro consists of two domains (D1 and D2) in the form of β-barrels connected by a flexible linker. The active site of NS3pro is formed by the triad His55–Asp88–Ser163(Gly161) located between the β-barrels. HSTV NS3pro complexed with the cytoplasmic domain of HSTV NSTR1 forms a closed conformation. This fact indicates that HSTV NSTR1 may be a cofactor of HSTV NS3pro, similar to *Orthoflaviviruses*. As previously shown using nuclear magnetic resonance, the complex of the NS2B cytoplasmic domain and NS3Pro of *Orthoflaviviruses* occurs primarily in a closed conformation, which provides valuable information on the conformational changes of proteases in the absence and presence of substrates and inhibitors that may be useful for the development of antiviral therapy [34]. The tertiary structure of HSTV NS3hel is typical of superfamily 2 helicases. HSTV NS3hel consists of three domains (D1–D3) and eight structural motifs (I, Ia, II, III, IV, IVa, V, and VI) situated in D1 and D2. For example, motifs I, Walker A and II, or Walker B, III, and VI are responsible for ATP binding and hydrolysis. The secondary structure of motif I forms a phosphate loop allowing for the residues within the motif to bind the β-phosphate of bound nucleotide triphosphate [35]. Motif II is involved in magnesium ion coordination in the ATP-binding pocket. Motif III is located near the ATP hydrolysis active site. Motif III coordinates with motifs I, II, and VI and forms an RNA-binding cleft [36]. Motif VI is known as the arginine finger and stabilizes interactions between the residues within the ATPase active site and the nucleic acid base of the bound nucleotide triphosphate molecule. Motifs Ia, IV, IVa, and V are responsible for interdomain interactions of NS3hel and binding of NS3hel to viral RNA [37]. The model of a produced biomolecular complex of HSTV NS3hel, ATP, and an HSTV RNA fragment shows that the viral RNA is indeed coordinated in a positively charged tunnel along the boundary of HSTV NS3hel D3 that directly interacts with HSTV NS3hel D1 and D2. This may indicate the correct functional annotation of NS3hel in the HSTV polyprotein.

RdRps encoded by RNA viruses are a unique class of nucleic acid polymerases. RdRps play a central role in viral genome replication and are therefore required for the viral life cycle in the host cell [38]. All RNA virus RdRps adopt a right-hand shape with palm, finger, and thumb domains surrounding the active site [39]. The tertiary structure of HSTV NS5RdRp has an encircled human right-hand architecture, typical of RdRps, and contains all structural motifs (A–F), despite an extremely low level of primary structure identity. Motifs A, B, C, and F directly interact with the NTP substrate and contain highly conserved residues. In contrast, motifs D, E, and G, located at the active site periphery, play primarily structural roles and are less conserved in their sequences [38]. Correct folding of motif E may affect the accuracy of the RdRp function because it is located near the priming loop [40]. In HSTV NS5RdRp, we identified two additional α21 and α22 extending from the putative priming loop of HSTV NS5RdRp. We suggest that α21 and α22 may promote priming loop stabilization in the enzyme active site. In addition, the spatial model for this region has a low pLDTT confidence score, which may be due to the fact that the priming loop structure should be specific to the RNA of a particular virus.

Many viral genomes encode MTase domains whose primary role is to methylate the 5′-terminal cap structures of viral RNAs for RNA degradation protection and efficient genome translation. Some viral MTase domains are identical to cellular FtsJ/RrmJ-like MTases involved in cellular RNA modification, whereas other MTases found in (+)-RNA viruses belong to a separate Sindbis-like family [41]. NS5MTases from *Orthoflavivirus* genus members adopt an α/β/α topology, form, together with RdRp, the NS5 protein, and are unique in that they simultaneously possess guanylyltransferase (GTase), N7 MTase, and 2′-O-MTase activities [42]. We identified NS5MTase in the HSTV polyprotein, which has a characteristic α/β/α structure. However, NS5MTase and NS5RdRp in HSTV are separated by an unknown NS5-X domain that is folded only in a complex with NS5MTase and NS5RdRp. This is highly unusual for *Orthoflavivirus* genus members. To date, two modes of functional conformations of *Orthoflavivirus* NS5 have been identified by X-ray crystallography. The conformation similar to that of Japanese encephalitis virus NS5 typically has a fully folded NS5RdRp finger domain stabilized by intramolecular interactions of the MTase–RdRp complex, and the NS5RdRp ring fingertip is involved in interfacial interactions. In contrast, the conformation similar to that of Dengue virus NS5 has the NS5RdRp finger domain stabilized by NS5MTase, but without the NS5RdRp ring fingertip [38]. We suggest that the unusual arrangement of HSTV NS5MTase relative to HSTV NS5RdRp may be an example of a novel functional conformation of NS5 from *Orthoflavivirus* genus members. However, this requires additional structural evidence, such as X-ray crystallography of the HSTV NS5MTase–NS5-X–NS5RdRp complex.

In this study, we have identified, for the first time, the putative nonstructural proteins of the recently discovered Haseki tick virus and found that their tertiary structure of proteins is similar to those of *Orthoflaviviruses*. At the same time, differences were found that are not typical for the *Flaviviridae* family. This suggests a hypothesis about the possible common origin of these viruses and evolutionary or taxonomic unity. But according to the phylogenetic trees presented in Kartashov’s article, HSTV-related viruses form a separate monophyletic group within the *Flaviviridae* family [4]. It is possible that HSTV has existed in the population for a long time, actively adapted to the environment, and may be the progenitor of *Orthoflaviviruses*. This explains why HSTV remained unnoticed for so long but requires proof.

Our results demonstrate the capabilities of the AlphaFold 3 neural network for annotation of non-homologous viral genomes and prediction of novel viral protein folds, giving up the secrets of recently discovered unclassified viruses. Of course, computational structural methods still require protein structure confirmation by experimental methods, but they already enable the rational design of proteins of novel viruses, increasing the likelihood of successful outcomes of structural experiments.

## 4. Materials and Methods

### 4.1. HSTV Sequence

To annotate HSTV nonstructural proteins, we used the whole-genome HSTV sequence under the accession number MW808978 from the NCBI (National Library of Medicine, Bethesda, MD, USA) GenBank (HSTV polyprotein sequence GenBank: UTQ11742).

### 4.2. Multiple Sequence Alignment (MSA) and Analysis

Closely related proteins with experimentally solved spatial structures were searched with NCBI BLAST at the Protein Data Bank (PDB) using the blastp (protein–protein BLAST) algorithm (accessed on 1 September 2024). Multiple amino acid sequence alignments were performed using Many-against-Many-searching (MMSeqs2) [43]. Only amino acid sequences of proteins with tertiary structures annotated in PDB were used for MSA. We used HMMER v3.1b2, (University of California, Santa Cruz, CA, USA), https://www.ebi.ac.uk/Tools/hmmer/ (accessed on 1 September 2024) [44] to search for protein domains in the Pfam database and NCBI Conserved Domains Database (CDD).

### 4.3. Search for Transmembrane Nonstructural Proteins

The search for transmembrane domains of nonstructural proteins in the polyprotein was performed using online services for membrane protein profile prediction: CCTOP version v1.1.0, (Protein Bioinformatics Research Group, Institute of Enzymology, RCNS, Budapest, Hungary), https://cctop.ttk.hu/ (accessed on 1 September 2024) [45] and TMHMM result version 2.0, (Department of Health Technology, Kgs. Lyngby, Denmark), https://services.healthtech.dtu.dk/services/TMHMM-2.0/ (accessed on 1 September 2024) [46].

### 4.4. Model Building Using AlphaFold 3 and Structural Alignments

Modeling of the spatial structures of putative HSTV proteins was performed using the AlphaFold 3 server (Google DeepMind, London, UK), https://alphafoldserver.com/welcome (accessed on 1 September 2024) [47]. The boundaries of putative HSTV nonstructural proteins in the polyprotein were determined by routine modeling of all possible protein structures in AlphaFold 3 and comparison of AlphaFold 3 structure models with all structures available in the PDB and AlphaFold databases using the FoldSeek server (Seoul National University, Seoul, South Korea), https://search.foldseek.com/ (accessed on 1 September 2024) [48]. Spatial models for further analysis were selected based on the confidence coefficient for each amino acid with allowance for an AlphaFold 3 predicted local distance difference (pLDDT) scaled from 0 to 100, which estimates the difference in Cα interatomic distances between the reference and the predicted structures. Pairwise alignment of the spatial structures of viral proteins and generated spatial structure models of HSTV proteins was performed using a Pairwise Structure Alignment tool (Research Collaboratory for Structural Bioinformatics Protein Data Bank (RCSB PDB), Piscataway, NJ, USA), https://www.rcsb.org/alignment/ (accessed on 1 September 2024) with TM-align for pairwise structural alignment [49]. The level of topological similarity was evaluated based on the root mean square deviation (RMSD) coefficient and assessment of the number of superimposed atoms in structures (TM score) on a scale from 0 to 1, where 1 indicates a perfect match between the predicted model and the reference structure. The alignment of secondary structures of HSTV viral proteins with those of *Flaviviridae* family members was visualized using the ESPript 3.0 software (Institute for the Biology and Chemistry of Proteins, Lyon, France), https://espript.ibcp.fr/ (accessed on 1 September 2024) [50]. Tertiary structure models of viral proteins were visualized using UCSF ChimeraX (Version 1.15rc) [51] and Mol*, (Protein Data Bank in Europe (PDBe), Hinxton, UK; Research Collaboratory for Structural Bioinformatics Protein Data Bank (RCSB PDB), Piscataway, NJ, USA; CEITEC - Central European Institute of Technology, Brno, Czech Republic; ELIXIR CZ, Praha, Czech Republic), https://molstar.org/ (accessed on 1 September 2024) [52] software version 4.10.0.

### 4.5. Protein Structure and Function Analysis

Hydrophobic regions of nonstructural protein models were generated using the ProteinTools server (University of Bayreuth, Bayreuth, Germany) in the Hydrophobic clusters mode, https://proteintools.uni-bayreuth.de/clusters/ (accessed on 1 September 2024) [53]. PyMol 3.0 (Schrodinger Sales Center, New York, NY, USA) with the Adaptive Poisson–Boltzmann Solver plugin (https://apbs.readthedocs.io/en/latest/index.html) (accessed on 1 September 2024) was used to calculate the electrostatic potential of generated nonstructural protein models. Topological diagrams and secondary structure diagrams of Haseki tick virus proteins were obtained using the PDB-sum service (European Bioinformatics Institute, Cambridge, UK), https://www.ebi.ac.uk/thornton-srv/software/PDBsum1/ (accessed on 1 September 2024).

## 5. Conclusions

In this study, we, for the first time, structurally annotated putative NS3 helicase, NS3 protease, NS5 methyltransferase, and NS5 RNA-dependent RNA polymerase of the novel Haseki tick virus, which have extremely low protein sequence identity coefficients compared with proteins from PDB. In addition, we proposed the arrangement of transmembrane proteins in the HSTV polyprotein and models of HSTV biomolecular complexes. The high structural alignment similarity of HSTV NS proteins to *Orthoflavivirus* NS proteins suggests a hypothesis of a possible common origin of these viruses and evolutionary or taxonomic unity. Our results provide insights into the genomic structure and evolution of the Haseki tick virus, which may become a health concern in the upcoming years as ticks expand their distribution range. Future studies will be aimed at confirming the tertiary structures of HSTV NS proteins by HSTV recombinant proteins’ production and synchrotron X-ray crystallography to gain fundamental knowledge about the structure of HSTV and a targeted approach to the development of POC tests and biologics.

## Figures and Tables

**Figure 1 ijms-25-13654-f001:**
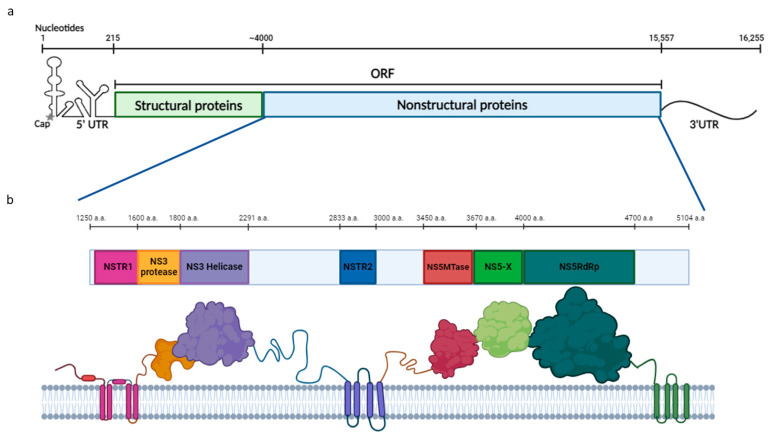
Haseki tick virus genome structure and chain topology of the translated single polyprotein. (**a**) Schematic representation of the complete HSTV genome and (**b**) nonstructural part of the HSTV genome with chain topology of the translated single polyprotein.

**Figure 2 ijms-25-13654-f002:**
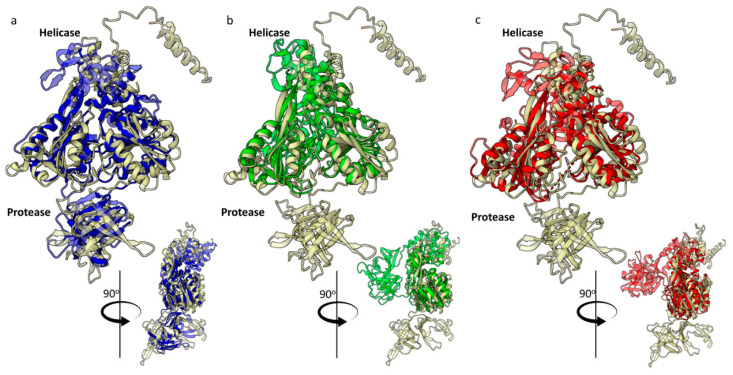
Imposition models of putative NS3 HSTV tertiary structures (ivory) with (**a**) dengue virus 4, PDB ID: 2VBC (blue), (**b**) hepatitis C virus, PDB ID: 2F9U (green), and (**c**) classical swine fever virus, PDB ID: 5WX1 (red).

**Figure 3 ijms-25-13654-f003:**
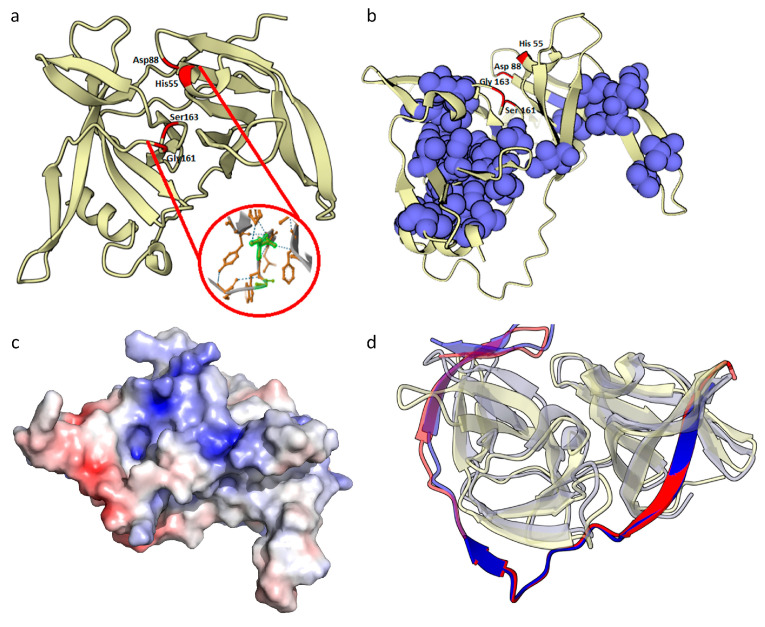
Putative HSTV NS3 protease tertiary structure: (**a**) catalytic site (red) and key amino acids of HSTV NS3pro (red circle), (**b**) hydrophobic clusters (blue) of HSTV NS3pro, (**c**) electrostatic surface potential of HSTV NS3pro, and (**d**) imposition models of spatial structure of HSTV NS3pro (ivory) in complex with NSTR1 extracellular domain (red) and Zika virus (PDB ID: 5H6V) NS3pro (grey) in complex with NS2B cofactor (blue). The positive surface potential is colored blue, and the negative surface potential is colored red.

**Figure 4 ijms-25-13654-f004:**
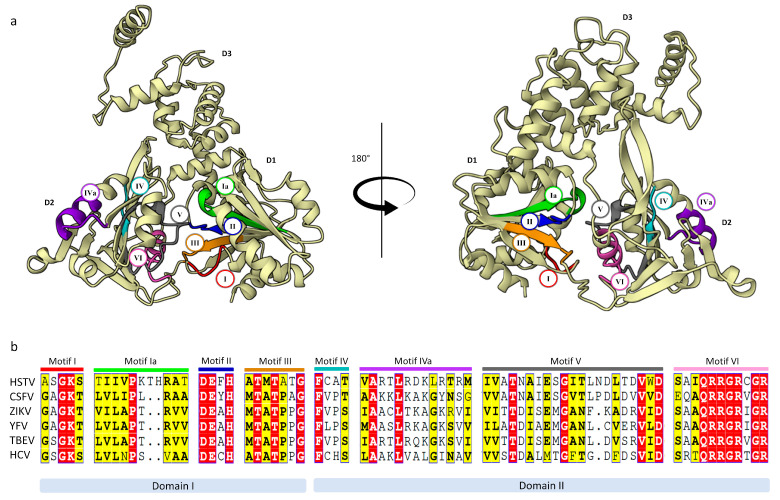
Functional regions of putative HSTV NS3hel. (**a**) The overall tertiary structure of HSTV NS3hel (ivory) with conservative motifs: motif I (red), motif Ia (green), motif II (blue), motif III (orange), motif IV (cyan), motif IVa (purple), motif V (grey), and motif VI (pink). (**b**) Sequence alignment of the conservative motifs: red boxes—100% aligned a.a. residues; yellow boxes—80% aligned a.a. residues; white boxes - unaligned. Abbreviations: CSFV—classical swine fever virus; ZIKV—Zika virus; YFV—yellow fever virus; TBEV—tick-borne encephalitis virus; HCV—hepatitis C virus.

**Figure 5 ijms-25-13654-f005:**
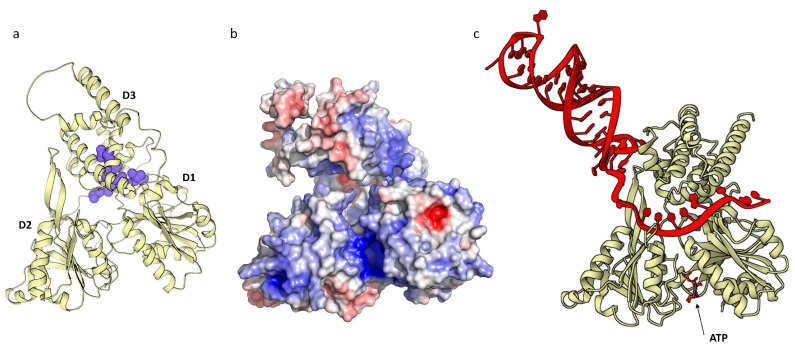
Putative NS3 helicase HSTV tertiary structure (ivory): (**a**) hydrophobic clusters (blue) of HSTV NS3hel, (**b**) electrostatic surface potential of HSTV NS3hel, and (**c**) tertiary structure of HSTV NS3hel (ivory) with RNA (red) and ATP. The positive surface potential is colored blue, and the negative surface potential is colored red.

**Figure 6 ijms-25-13654-f006:**
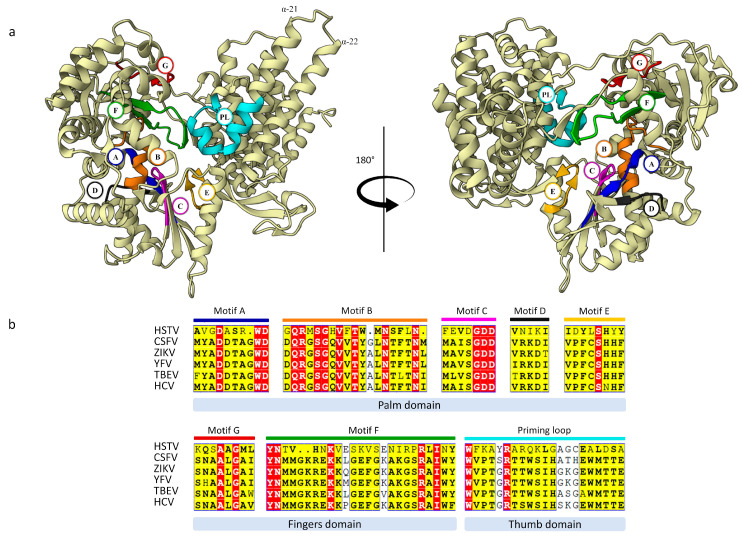
Functional regions of putative HSTV NS5RdRp. (**a**) The overall tertiary structure of HSTV NS5RdRp (ivory) with catalytic motifs: motif A (blue), motif B (orange), motif C (magenta), motif D (black), motif E (yellow), motif F (green), motif G (red), and priming loop (PL) (cyan). (**b**) Sequence alignment of the HSTV NS5RdRp motifs: red boxes—100% aligned a.a. residues, yellow boxes—80% aligned a.a. residues, white boxes—unaligned. Abbreviations: CSFV—classical swine fever virus; ZIKV—Zika virus; YFV—yellow fever virus; TBEV—tick-borne encephalitis virus; HCV—hepatitis C virus.

**Figure 7 ijms-25-13654-f007:**
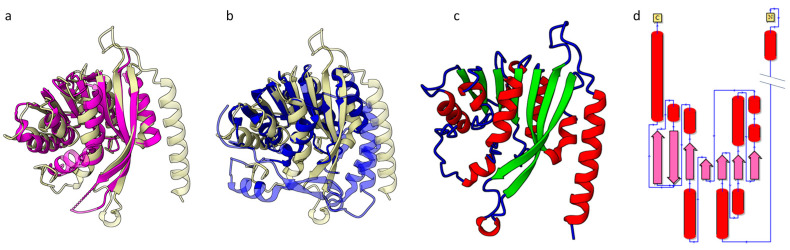
HSTV NS5Mtase structure. (**a**,**b**) Imposition models of HSTV NS5MTase spatial structure (ivory) with (**a**) NS5Mtase *Pyrococcus horikoshii*, PDB ID: 1WY7 (magenta); (**b**) NS5Mtase Dengue virus 3, PDB ID: 3P97 (blue); (**c**) the model of HSTV NS5MTase spatial structure: α-helix (red), β-strand (green); and (**d**) topology diagram of HSTV NS5MTase: α-helix (red), β-strands (pink), N-amino-terminus, C-carboxyl-terminus.

**Figure 8 ijms-25-13654-f008:**
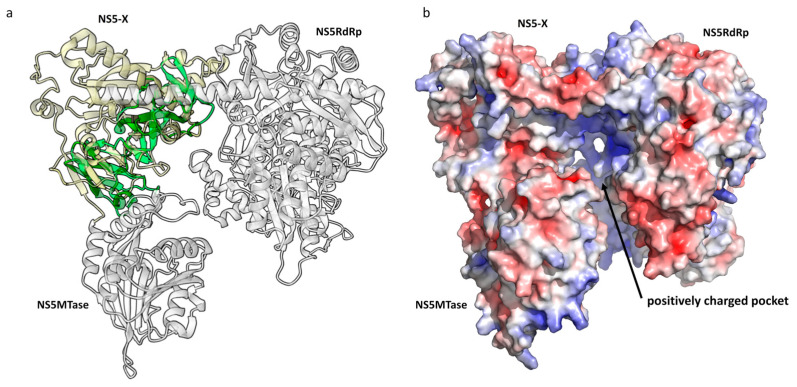
Spatial structure of putative HSTV NS5Mtase in complex with HSTV NS5-X and HSTV NS5RdRp. (**a**) Imposition model of HSTV NS5MTase–NS5-X–NS5RdRpspatial structure (gray and ivory) with NS5A zinc-binding domain of hepatitis C virus (green) (PDB ID: 1ZH1) and (**b**) electrostatic surface potential of HSTV NS5Mtase-NS5-X-NS5RdRp. The positive surface potential is colored blue, and the negative surface potential is colored red.

**Table 1 ijms-25-13654-t001:** Comparison of amino acid sequences and tertiary structure of putative HSTV NS3 domains with NS3 domains of *Flaviviridae* family viruses.

PDB ID	Name of Virus	TM Score	RMSD, Å	Aligned Residues, a.a.	Amino Acid Sequence Identity ^1^, %
NS3pro HSTV					
2FOM	Dengue virus 2	0.79	2.33	137	23
2F9U	Hepatitis C virus	0.69	4.64	118	17
5WX1	Classical swine fever virus	0.60	3.22	138	22
NS3hel HSTV					
1A1V	Hepatitis C virus	0.66	3.46	328	20
2JLS	Dengue virus 4	0.63	4.38	310	14
7NXU	Tick-borne encephalitis virus	0.62	4.63	285	18
4CBL	Classical swine fever virus	0.63	4.45	283	18
D1-D2 NS3hel HSTV					
1A1V	Hepatitis C virus	0.76	2.45	268	21
2JLS	Dengue virus 4	0.74	3.26	258	17
7NXU	Tick-borne encephalitis virus	0.70	3.34	250	21
4CBL	Classical swine fever virus	0.61	05.02	183	15
D3 NS3hel HSTV					
1A1V	Hepatitis C virus	0.30	6.16	47	5
2JLS	Dengue virus 4	0.36	5.7	57	5
7NXU	Tick-borne encephalitis virus	0.40	5.40	67	3
4CBL	Classical swine fever virus	0.39	5.79	59	8

^1^ Amino acid sequence identity was calculated relative to the aligned residues in the tertiary structure.

**Table 2 ijms-25-13654-t002:** Comparison of amino acid sequences and tertiary structure of putative NS5RdRp and NS5MTase of HSTV with NS5RdRp and NS5MTase of viruses of the *Flaviviridae* family.

PDB ID	Name of Virus	TM Score	RMSD, Å	Aligned Residues, a.a.	^1^ Amino Acid Sequence Identity, %
NS5RdRp					
7XD8	Dengue virus 2	0.72	3.26	498	13
7D6N	Tick-borne encephalitis virus	0.69	3.50	473	12
7EKJ	Classical swine fever virus	0.66	3.54	476	18
6GP9	Hepatitis C virus	0.63	3.85	431	14
NS5MTase					
1WY7	*Pyrococcus horikoshii*	0.77	3.02	164	22
3P97	Dengue virus 2	0.60	4.18	142	14
7D6M	Tick-borne encephalitis virus	0.60	4.21	143	11
2WA1	Modoc virus	0.59	3.69	142	12
8GY4	Alongshan virus	0.57	3.89	140	10

^1^ Amino acid sequence identity was calculated relative to the aligned residues in the tertiary structure.

## Data Availability

The original contributions presented in the study are included in the article/Appendix A; further inquiries can be directed to the corresponding author.

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
