# Peer review of "Tertiary Structures of Haseki Tick Virus Nonstructural Proteins Are Similar to Those of Orthoflaviviruses"

_ijms, 2024, doi:10.3390/ijms252413654_

Round 1

Reviewer 1 Report

Comments and Suggestions for Authors

This manuscript reports studies on predicted proteins of the recently discovered Haseki tick virus (HSTV) using alphafold 3. The manuscript is interesting, but needs significant revisions prior to publication.

Major issues are:

1.       All the protein structures are predicted, not proven. This needs to be made clear throughout the text., e.g. “putative” proteins

2.       There are no comparisons made with nonstructural proteins of tick-borne viruses. For example, tick borne encephalitis (TBE). The authors could argue there are no structures available for comparison but would not alphafold 3 predicted structures for TBE virus proteins with equivalent HSV predicted proteins be of interest, rather than the focus on mosquito-borne orthoflaviviruses.

3.       There are structures for dengue NS4A (Li et al. 2018; PMID: 29055659) and NS4B (Li et al 2016b; PMID: 27554985)

4.       An interesting point would be to compare the amino acid sequences in-between and N- and C- termini of predicted proteins to investigate putative cleavage sites to support the predicted proteins.

5.       The start of the Results section needs some information here on the genome organization. Maybe take the genome organization from reference 5 and update as a figure for the paper and mark on genome positions and predicted proteins, plus non-coding regions. This would help orientate readers at the start of the paper.

6.       Please do not use the term”‘dengue virus”. There are four dengue viruses and readers will need to know which dengue virus is being referred.

Minor points

Line 31: The ICTV has just published its 2023 report in Arch Virology.

Line 99: Replace “elucidate” with “Investigate” as no structure has been determined for any HSTV proteins.

Line 137: “orthoflavivirus”

Lines 407-408: It is incorrect to say there are no licensed dengue vaccines. Two are licensed in multiple countries.

Line 447: How can there by a “fact” with no structure information for HTSV proteins?

Line 540: “functionally annotated”. This is incorrect as all the structures are predictions and none proven so the functions cannot be annotated as no functional assays have been undertaken.

Author Response

Thank you for reviewing our manuscript (ijms-3354469) entitled “Tertiary structures of Haseki tick virus nonstructural proteins are similar to those of Orthoflaviviruses” submitted for publication in IJMS. 

We thank you for valuable suggestions that allowed us to make the manuscript more convincing and understandable. We accepted your suggestion and made corresponding change in the manuscript. We major revised our article.

Below please find our detailed responses to your questions and comments. All modifications in the manuscript have been highlighted in red.

Reviewer 2 Report

Comments and Suggestions for Authors

Abstract

o Highlight the potential applications of the findings more prominently (e.g., antiviral development).

o The mention of AlphaFold 3 is appropriate, but clarify its novel application compared to prior studies.

Introduction

o It discusses global viral challenges but should better specify why Haseki tick virus (HSTV) is a critical addition to the research field.

o While implicit, explicitly state the hypothesis and the study's novelty compared to existing Flaviviridae research.

Results

o Discuss why structural divergence from known Flaviviridae members is significant.

Discussion

o Discuss the potential evolutionary trajectory of HSTV within Flaviviridae more extensively.

Conclusion

o Restate the study’s significance succinctly while avoiding redundancy from the discussion.

o Outline specific next steps, such as X-ray crystallography, to confirm computational predictions.

Author Response

(The authors gave the same response as above.)

Round 2

Reviewer 1 Report

Comments and Suggestions for Authors

All of my comments have been addressed except for:

Line 108: Change to “yellow fever, Japanese encephalitis and tick-borne encephalitis vaccines”

Lins 456-457: This needs correcting again, as it is wrong to say there are no effective dengue vaccines. Reference 31 is a poor choice and is wrong. THERE ARE TWO LICENSED DENGUE VACCINES …. DENGVAXIA (SANOFI) AND QDENGA (TAKEDA)

Author Response

Dear Reviewer, thank you for reviewing our manuscript. We accepted your suggestion and made corresponding change in the manuscript. Below please find our detailed responses to your questions and comments. All modifications in the manuscript have been highlighted in red.
